# Ichthyological Differentiation and Homogenization in the Pánuco Basin, Mexico

**Norma Martínez-Lendech** [1], **Ana P. Martínez-Falcón** [2], **Juan Jacobo Schmitter-Soto** [3], **Humberto Mejía-Mojica** [4], **Valentino Sorani-Dalbón** [1], **Gabriel I. Cruz-Ruíz** [5] and **Norman Mercado-Silva** [1,*]

1  Centro de Investigación en Biodiversidad y Conservación, Universidad Autónoma del Estado de Morelos, Cuernavaca Morelos 62209, Mexico; norma.mlendech@uaem.edu.mx (N.M.-L.); vsorani@uaem.mx (V.S.-D.)
2  Centro de Investigaciones Biológicas, Universidad Autónoma del Estado de Hidalgo, Pachuca Hidalgo 42001, Mexico; apmartinez@cieco.unam.mx
3  Departamento de Sistemática y Ecología Acuática, El Colegio de la Frontera Sur, Chetumal Quintana Roo 77010, Mexico; jschmitt@ecosur.mx
4  Centro de Investigaciones Biológicas, Universidad Autónoma del Estado de Morelos, Cuernavaca Morelos 62209, Mexico; humberto@uaem.mx
5  Centro Interdisciplinario de Investigación para el Desarrollo Integral Regional, Unidad Oaxaca, Instituto Politécnico Nacional, Santa Cruz Xococotlán Oaxaca 71230, Mexico; gicruzr@ipn.mx
*  Correspondence: norman.mercado@uaem.mx

**Abstract:** Species introductions and extirpations are key aspects of aquatic ecosystem change that need to be examined at large geographic and temporal scales. The Pánuco Basin (Eastern Mexico) has high ichthyological diversity and ecological heterogeneity. However, freshwater fish (FWF) introductions and extirpations since the mid-1900s have modified species range and distribution. We examine changes in FWF species composition in and among four sub-basins of the Pánuco by comparing fish collection records pre-1980 to 2018. Currently, the FWF of the Pánuco includes 95 species. Fishes in the Poeciliidae, Cyprinidae, and Cichlidae, respectively, comprised most records over time. Significant differences in species composition were found between the first (pre-1980) and last (2011–2018) study periods, but not for periods in-between. Eight independent species groups were key for explaining changes in Pánuco river ichthyofauna; one group was dominated by invasive species, and saw increases in the number of records across study periods (faunal homogenization). Another group was formed by species with conservation concern with a declining number of records over time. Thirteen (2 native and 11 non-native) species were responsible for temporal turnover. These results strongly suggest high rates of differentiation over time (via native species loss) following widespread non-native species introductions.

**Keywords:** biogeography; community composition; differentiation; freshwater fishes; homogenization; Pánuco basin; watershed

## 1. Introduction

Rapid ecosystem and species loss results from pollution, as well as land use and climate change [1]. Human activity does not solely lead to species loss; it can also lead to increases in faunal similarity by the alteration of species range [2] via species introductions and species loss. Anthropogenic introductions expand a species´ range beyond its natural dispersal capacity; species loss can result from human driven ecosystem and habitat deterioration [2–5]. The number and manner of species loss and introductions occurring can result in different levels of biological homogenization and differentiation [5,6]. Lacking extirpations, the introduction of a given invasive in two sites leads to an

increase in their similarity. Alternatively, the introduction of different invasive species in two sites will reduce among-site similarity [5]. Loss of species shared among two sites will also generate biological differentiation [5,7]. Human-caused biological homogenization and differentiation are serious threats to global biodiversity [8–10]. Trends in both biological homogenization and differentiation can be used as tools for the development of conservation strategies [7]. Processes of homogenization and differentiation have been documented for freshwater fish faunas in North America and Europe [8–11]. Generally, studies have found increasing rates of among-region homogenization caused primarily by non-native species introductions and translocations, as well as urbanization, but also loss of community similarity driven by non-native species in certain settings [10]. While the expansion and disappearance of non-native and native species, respectively, have been the focus of an increasing body of literature in Latin America [12–14], processes of homogenization or differentiation have not been formally addressed in published literature in Mexico. Understanding the rates and causes of homogenization and differentiation in this species-rich area of the world could help identify strategies to help in conservation efforts of unique faunas. As a first step in understanding whether and how these processes are happening in the freshwater fish fauna of the Pánuco Basin, here, we examine their rates in one of the most important basins of Mexico.

The Pánuco Basin is one of the largest (9.7 million ha) and most diverse in the Mexican Gulf of Mexico slope [15]. Nearctic and Neotropical freshwater fish (FWF) faunas converge in this basin, which is also a transition zone between three Mexican physiographic regions [16,17]. A complex geology has led to a relatively high proportion (57%) of FWF in Mexico being endemic or microendemic [18]. Approximately 95 (native and non-native) species have been recorded in the basin [19] from a variety of sources, including species lists and inventories, and conservation-oriented studies [17,20–22]. Some studies have provided evidence on the impacts of introduced species on natives such as *Cualac tessellatus, Ataeniobius toweri, Tampichthys mandibularis,* and *T. dichroma* [21,23] in the Pánuco. However, homogenization and differentiation processes have not been addressed for the basin. The analysis of these processes can shed light on the temporal dynamics and causes that have led to the current distribution of species in the Pánuco. Here, we explore and quantify FWF homogenization and differentiation processes for four sub-basins in the Pánuco based on fish collections and museum information spanning more than 50 years of data. We hypothesize a temporal increase in FWF homogenization and then differentiation for the entire Pánuco, as well as among the four sub-basins.

## 2. Materials and Methods

### 2.1. Units of Analysis

The Pánuco basin spans a large (97,196 km$^2$), ecologically and geographically diverse area. The mainstem of the Pánuco runs generally SW–NE for 510 km from its source to the Gulf of Mexico [24]. The Pánuco runs from the central Mexican plateau and the Sierra Madre Oriental, along rugged terrain to reach a relatively short coastal plateau. It drains the states of Tamaulipas, San Luis Potosí, Hidalgo, Querétaro, México, Guanajuato, and Veracruz [7] (Figure 1). As classified by the Mexican Institute for Geography and Statistics (INEGI), the National Institute of Ecology (INE) and the National Water Commission (CONAGUA), the Pánuco includes four sub-basins :(A) Río Pánuco, (B) Río Tamesí, (C) Río Tamuín, and (D) Río Moctezuma [25] (Figure 1). The Río Pánuco sub-basin is in the states of Tamaulipas and Veracruz [16]; it is the lowest portion of the Pánuco Basin and is comprised of the outlet of the San Juan and Tula rivers downstream from the Zimapán Dam on the Moctezuma and the confluence of the Tamuín River. The Rio Tamesí (originates with the name of Guayalejo river) sub-basin flows NW–SE in the states of Tamaulipas, San Luis Potosí, and Veracruz [7,24]. The Río Tamuín starts in the state of San Luis Potosí after the confluence of the Verde and Santa María rivers, and includes the Gallinas, Tamasopo, El Salto, and Valles rivers. The Calabazas and Los Hules rivers are also located in this sub-basin. The Moctezuma sub-basin includes the San Juan, Extoraz (Tolimán and Victoria), Amajac, and Tempoal rivers and drains the states of Hidalgo, Querétaro, México, San

Luis Potosí, Guanajuato, and Veracruz [7]. Administratively, this sub-basin includes the once endorheic Mexico City valley as part of the Moctezuma; this valley has not been included in our study.

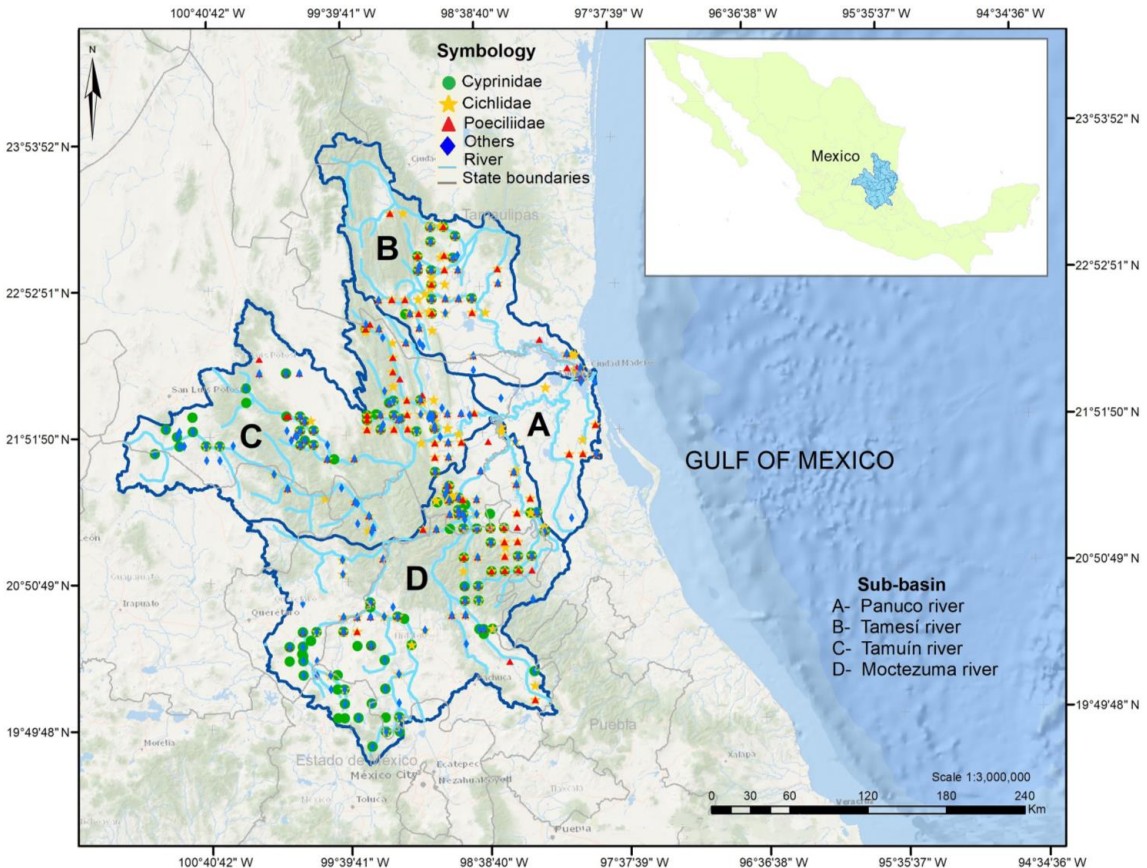

**Figure 1.** The Pánuco Basin, in Eastern central Mexico, with four sub-basins (sensu INEGI, 2007). Map includes locations where records for the most diverse families, Cyprinidae, Cichlidae, and Poeciliidae, were obtained. "Others" refers to other families found in each location. Coordinates in universal transverse Mercator (UTM) units for zone 14 N.

The above sub-basins were used as geographical units of analysis upon which we studied homogenization and differentiation processes. That is, FWF changes though time were analyzed in each sub-basin and contrasted with those of other sub-basins.

*2.2. Database*

Sub-basin specific databases were constructed using historical (pre-1980 to 2018) FWF collection information. The presence of only primary and secondary species (FWF with little or no tolerance to ocean salinity) [26] collected in sites located in each of the Pánuco sub-basins was included in the database. Databases were populated primarily from data in FISHNET2 [27] and the Global Biodiversity Information Facility [28]. Both repositories were searched using Pánuco and Panuco as geographic keywords to locate records for the river basin. GBIF data included a 2018 update from the Colección Nacional de Peces Dulceacuícolas Mexicanos de la Escuela Nacional de Ciencias Biológicas [19]. The geographic location for each record found in databases was verified using QGIS 3.8 and adjusted, when necessary, to fall within each sub-basin. Fish records from Miller et al. (2009) [15], from the "Edmundo Díaz Pardo" fish collection at the Universidad Autónoma de Querétaro, were also obtained and their coordinates were revised. Additionally, the Web of Science (Clarivate Analytics©, Philadelphia, PA, USA.) was searched for recent (up to 18 years prior to 2018) publications reporting on Pánuco river

collections or other ichthyological studies [29–62]. Species and genera names in each collection or source were verified and updated [63,64] to adjust synonymy and eliminate possible misidentifications.

Using its date, each collection was assigned to one of five time intervals. Interval 1 included pre-1980 (including 1980) records; interval 2 included records from between 1981 and 1990; and intervals 3, 4, and 5 spanned 1991–2000, 2001–2010, and 2011–2018, respectively.

Thus, the resulting database had presence-absence (0,1) data for each species, an identifier for the sub-basin where it was collected (A, B, C, D) and the time interval (1–5) when it was recorded. These data were used to carry out similarity analyses described below.

We defined species native to the Pánuco as those whose original distribution included a waterbody in the Pánuco. Any species with natural range outside the Pánuco was regarded as non-native. Among these, invasive species were defined based on the categorization by the Comisión Nacional para el Conocimiento y Uso de la Biodiversidad (CONABIO), which defines invasive species for Mexico [65].

*2.3. Analyses*

We first estimated sample completeness [66], using species accumulation curves (bootstrap) from all intervals with EstimateS [67]. We then quantified biological homogenization or differentiation between sub-basins and among time intervals. We used Jaccard´s similarity index to calculate distance matrices among species and time intervals, and among sub-basins using PERMANOVA (permutational multivariate analysis of variance) [68,69]. Next, we carried out a pairwise comparison among intervals and sub-basins, graphically displaying Jaccard values via a boostrapped MDS (multidimensional scaling) procedure among intervals and sub-basins. PERMANOVA and MDS were performed in Primer 7 [70]. Subsequently, we carried out a dual cluster analysis to identify species associations throughout the Pánuco, based on the time interval and the sub-basin where they were found. To achieve this, we converted presence-absence (0,1) data into ordinal data (i.e., the number of sub-basins in which a species is present per time interval) and used Ward´s method for calculating Euclidean distances among time-intervals and sub-basins [71] in Past3 [72].

To identify species responsible for among sub-basin and time-period differences, we carried out four generalized discriminant analyses (GDA). Two (one for sub-basin differences and one for time-period differences) included all 95 species; they identified 15 species with the highest correlations. Two more GDAs (one for sub-basin differences and one for time-period differences) were carried out using only these 15 species. These analyses were carried out in software Statistica v.10 [73].

## 3. Results

Our database included 95 species in 35 genera, 12 families, and 10 orders (Figure 1). Families with the most species were Poeciliidae (34.7%), Cyprinidae (20%), and Cichlidae (15.8%) (Tables A1 and A2). Species accumulation curves for all intervals resulted in 94.79% of estimated richness. For each interval, independently, species accumulation curves represented >80% of estimated richness.

Significant differences in species composition were detected among the oldest and most recent time intervals according to pairwise comparison procedures ($t = 1.5705$, $p = 0.022$; Figure 2a); however, no differences were found among other time intervals. Species composition differed among all sub-basins (Fpseudo = 9.2383, df = 3, $p = 0.001$; Figure 2b). The Río Pánuco and the Río Tamuín sub-basins shared the least species (24%) when all time intervals were considered. The Río Moctezuma and Río Tamuín sub-basins shared the most (55%) species. When comparing among intervals, intervals 2 and 3 (1981–1990 and 1990–2000, respectively) shared 91% of species. Intervals 1 and 5 (pre-1980 and 2011-2018, respectively) shared the least species (44%).

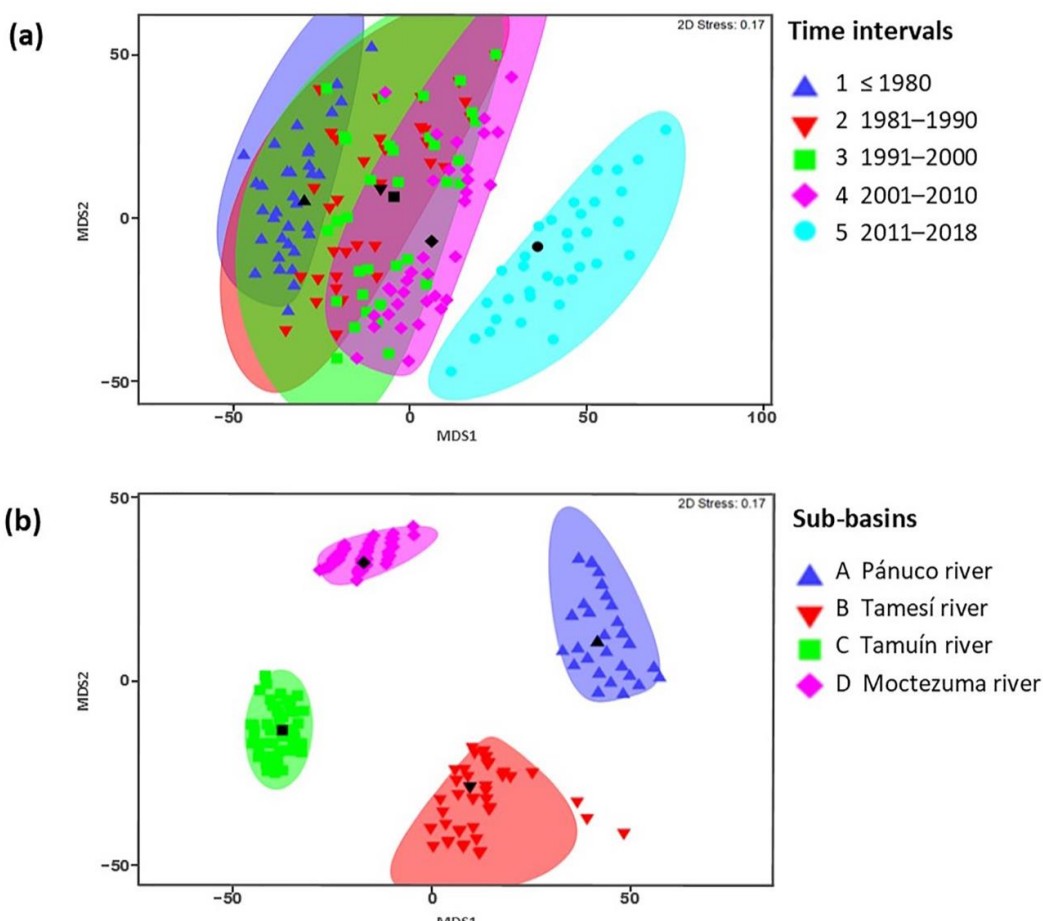

**Figure 2.** Multidimensional scaling (MDS) comparison (Jaccard values of similarity) of Pánuco River ichthyofauna between (**a**) time intervals and (**b**) sub-basins (see methods for details). Icons in black represent the centroids for each group with a similar shape.

Dual grouping analyses resulted in eight distinct groups. Group 1 had species found in more than two sub-basins in the first four intervals and whose presence in the last interval declined. Group 2 was formed by species present in more than three sub-basins in the first three time intervals and then declined after the fourth time interval. Fish species in group 3 were rare in the first intervals and were found in more than three sub-basins by the fourth and fifth time intervals. All fish species in group 4 were present in more than three sub-basins in all intervals. Group 5 was formed by species exclusive to each sub-basin, which were present in the first three intervals, but absent in the last one. Fishes in group 6 were present throughout all time intervals. Group 7 was formed by fish species present in the last three intervals only. Finally, group 8 included species recorded in time interval 1, but absent thereafter (Figure 3; Table 1).

The GDA analysis for time intervals explained 97.89% of the variation with two discriminant functions, DF1 (first discriminant function) 92.79% and DF2 (second discriminant function) 5.1%. The species that most contributed to the over time-species changes were *Algansea tincella, Amatitlania nigrofasciata, Ataeniobius toweri, Carassius auratus, Chirostoma grandocule, Coptodon rendalli, C. zillii, Ctenopharyngodon idella, Cyprinus carpio, Gambusia affinis, G. marshi, Ictiobus bubalus*, and *Notropis boucardi* (Table 2). For the sub-basin GDA analysis, we found 97.38% of the variation explained by two functions (DF1: 87.67% and DF2: 9.71%), highly correlated to the presence of *A. tincella, A. nigrofasciata, A. toweri, C. auratus, C. grandocule, C. rendalli, C. zillii, C. idella, C. carpio, G. marshi, Gambusia atrora, G. panuco, I. bubalus*, and *N. boucardi* (Table 3).

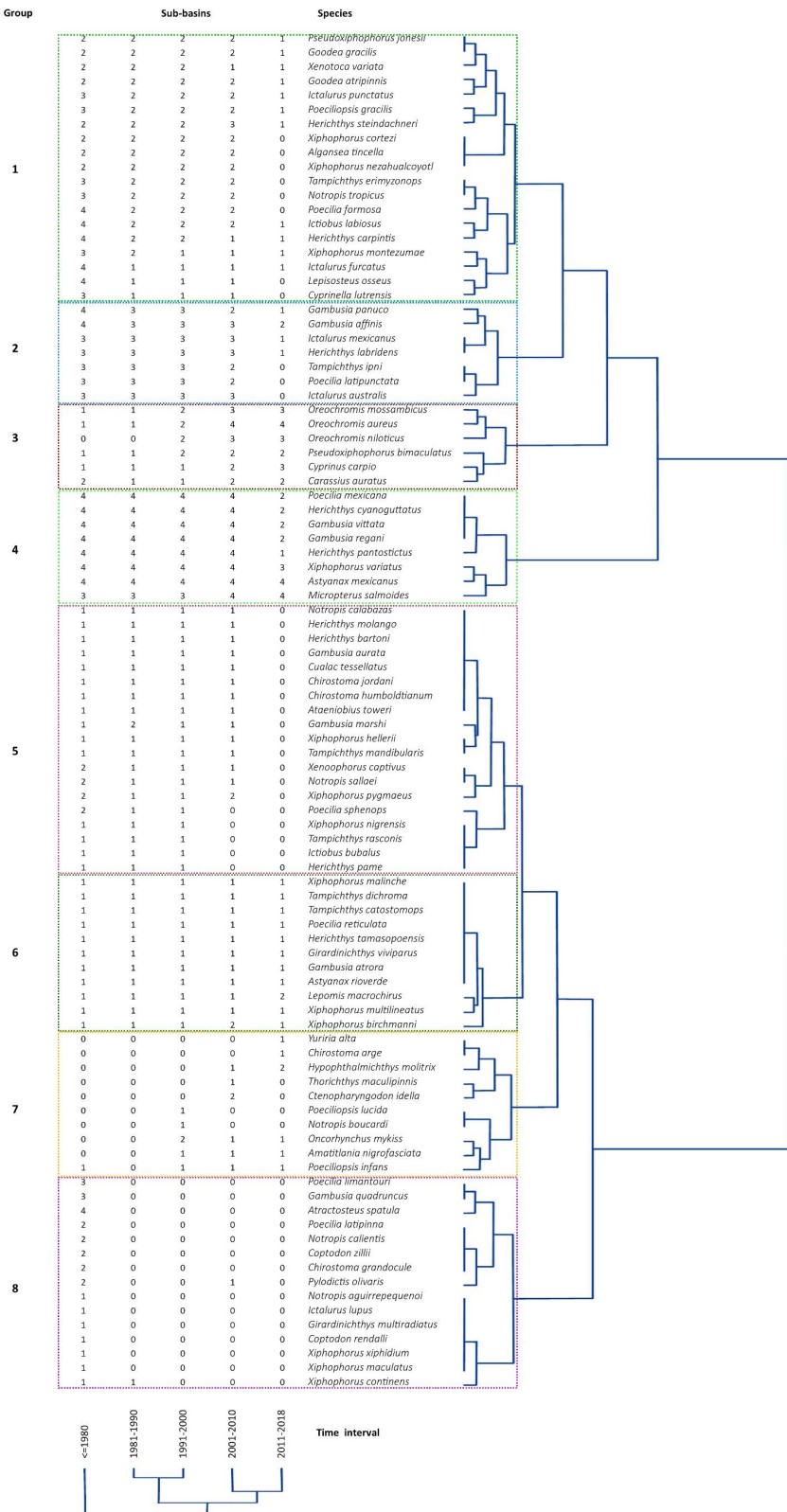

**Figure 3.** Time interval and sub-basin dual clustering analysis results for the Pánuco River ichthyofauna. The numbers from 0 to 4 in the columns indicate the number of sub-basins where a species was recorded. Groups formed in analyses are indicated by a number to the left of each colored box. The tree to the right indicates similarity in distribution among species and species groups. The tree at the bottom of the image indicates similarity among time intervals.

**Table 1.** Attributes of Pánuco River Basin fish species groups formed in cluster analyses. For each group, we present the number of species in the group; the number of natives, non-natives, and invasives; the number of species in each group categorized by IUCN or NOM 059 SEMARNAT 2010; and general characteristics for the group.

| Group | Species in Group | Native | Non Native | Invasive | IUCN * | NOM 059 SEMARNAT 2010 ** | Group Attributes |
|---|---|---|---|---|---|---|---|
| 1 | 19 | 17 | 2 | 0 | 2 | 2 | Eight species were not found in later time intervals. |
| 2 | 7 | 6 | 1 | 0 | 2 | 4 | Three species were not found in later time intervals. |
| 3 | 6 | 0 | 6 | 5 | 2 | 0 | All species present through time intervals. |
| 4 | 8 | 6 | 2 | 1 | 0 | 0 | All species present through time intervals. |
| 5 | 19 | 13 | 6 | 0 | 7 | 8 | Species were lost between time intervals 4 and 5. |
| 6 | 11 | 8 | 3 | 1 | 3 | 2 | All species present through time intervals. |
| 7 | 10 | 0 | 10 | 4 | 3 | 2 | Nine species were introduced after time interval 3, but four were not found in the last time interval. |
| 8 | 15 | 4 | 11 | 1 | 3 | 1 | Species were not reported after time interval 1. |

* International Union for Conservation of Nature (IUCN). Species categories: NT = near threatened, VU = vulnerable, EN = endangered, CR = critically endangered. ** NORMA Oficial Mexicana NOM-059-SEMARNAT-2010. Species categories: Pr = subject to special protection, A = threatened, P = in danger of extinction.

**Table 2.** Fish species from the Pánuco Basin correlated with discriminant function analyses for time intervals (see text). Eigenvalues and % of variance explained in a function are shown. DF: Discriminant function.

| Species | DF1 | DF2 |
|---|---|---|
| *Algansea tincella* | 0.038516 | 0.030990 |
| *Amatitlania nigrofasciata* | −0.006545 | −0.114626 |
| *Ataeniobius toweri* | 0.022237 | 0.017892 |
| *Carassius auratus* | −0.007613 | 0.028727 |
| *Chirostoma grandocule* | 0.008891 | 0.310992 |
| *Coptodon rendalli* | 0.005133 | 0.179551 |
| *Coptodon zillii* | 0.008891 | 0.310992 |
| *Ctenopharyngodon idella* | 0.035840 | −0.127134 |
| *Cyprinus carpio* | −0.029556 | −0.062774 |
| *Gambusia affinis* | 0.023831 | 0.103444 |
| *Gambusia marshi* | 0.024345 | 0.026311 |
| *Ictiobus bubalus* | 0.013731 | 0.063038 |
| *Notropis boucardi* | 0.012446 | −0.089352 |
| Eigenvalue | 126.94 | 7.05 |
| % of variance | 92.79 | 5.1 |

**Table 3.** Fish species from the Pánuco Basin correlated with discriminant function analyses for sub-basins (see text). Eigenvalues and % of variance explained in each function are shown. DF: Discriminant function.

| Species | DF1 | DF2 |
|---|---|---|
| *Algansea tincella* | 0.103136 | −0.044257 |
| *Amatitlania nigrofasciata* | 0.056162 | 0.165854 |
| *Ataeniobius toweri* | 0.054143 | −0.333427 |
| *Carassius auratus* | 0.080127 | 0.099953 |
| *Chirostoma grandocule* | 0.001050 | 0.068697 |
| *Coptodon rendalli* | 0.022928 | 0.067710 |
| *Coptodon zillii* | 0.005592 | 0.038124 |
| *Ctenopharyngodon idella* | −0.001050 | −0.068697 |
| *Cyprinus carpio* | 0.062067 | 0.143948 |
| *Gambusia atrora* | 0.049687 | 0.212331 |
| *Gambusia marshi* | −0.023292 | 0.029809 |
| *Gambusia panuco* | 0.029366 | −0.127961 |
| *Ictiobus bubalus* | −0.010438 | −0.070169 |
| *Notropis boucardi* | 0.013536 | −0.083357 |
| Eigenvalue | 179.29 | 19.86 |
| % of variance | 87.67 | 9.7 |

## 4. Discussion

### 4.1. Compositional Changes

Our study discovered differentiating trends in FWF fauna of the Pánuco River over the last 50 years. While significant differences were found only between the first and last interval, changes occurred gradually over our study period. The first and second time intervals shared 82% of species, with 15 species recorded in time period 1 being lost in interval 2. These included 12 non-native species (i.e., invasive *C. zilli* [65]), which were perhaps unable to become established, and three native species (i.e., critically endangered *Notropis calientis* [74]). Between time intervals 2 and 3, Pánuco sub-basins were more homogenous (similarity at 91%), and six non-native species were introduced (including invasives *A. nigrofasciata, Oncorhynchus mykiss,* and *O. niloticus,* [65]). Native species like *Xiphophorus continens* (data deficient according to [75]) were not found after the third time interval. Interestingly, species homogenization was only found in these first three time intervals, coinciding with a nation-wide policy for species introductions for aquaculture. During the 1970s and 1980s, several governmental programs promoted fish farms and planting of non-natives in newly created reservoirs [76]. These policies led to a substantial increase in the number of introduced species throughout the country [23,77,78]. Past these time intervals, we noted a decline in species similarity, which has continued to the present. The third and fourth intervals shared 86% of species, but four non-natives (including invasives *Hypophthalmichthys molitrix* and *C. idella* [65]) were first found in the Pánuco. Seven species, including *Herichthys pame, Xiphophorus nigrensis,* and *Tampichthys rasconis* (this last one being endangered [79]) were not found after interval 4. The highest differentiation (59% of species shared) was found between intervals 4 and 5, with 29% of species in period 4 not found in period 5, and two newly introduced species.

Gradual differentiation over time ultimately resulted in only 34% of species being shared between the first and last time intervals. This included 47% of species registered in the first time interval not being recorded in the latest time intervals and the introduction of 10% of nonnative species, of which 50% are considered invasive [65]. While we expected continually increasing homogenization over time in the Pánuco basin, what we detected was a marked increase in ichthyological differentiation (vía species loss) occurring after non-natives had been introduced in the region. Several studies have demonstrated that species invasions can result in native species loss [80–82]. Nile perch (*Lates niloticus*) introductions into Lake Victoria (Africa) [83] and loricariid introductions into Infiernillo Reservoir

in Mexico [84,85] are examples of how invasives have led to collapses of native faunas. There are, however, many instances in which it is difficult to establish a cause and effect relationship between a species introduction and the demise of native species. For example, the introduction of *Oreochromis mossambicus* in the State of Morelos in the 1970s coincided with the local extirpation of *Poeciliopsis balsas*. While the effects of the non-native species were cited as a likely cause for the extirpation [77,86], the mechanism leading to the disappearance of the native species was not clearly established. Similar to this study, in our analysis, we identified increases in the distribution of non-natives broadly coinciding with diminishing records for native species.

### 4.2. Assemblage Change among Time Intervals

Our cluster analyses helped us identify which species entered or were lost from sub-basins through time. Group 3 was integrated by invasive species that expanded their range after being introduced into one of the sub-basins. Group 7, also integrated by non-native species, was composed of species that did not appear in later time intervals, suggesting they were not able to become established. These two groups include 17% of all species found in the Pánuco Basin, and are species highly utilized in aquaculture, commercial and sport fisheries, and for biocontrol worldwide [78,86,87]. Groups 1 and 5 were mostly composed of species under conservation status and that were not found in later time intervals. These can be considered the most vulnerable groups and comprise 40% of species of the Pánuco. These groups include *Herichthys steindachneri*, *A. toweri*, *Herichthys bartoni*, *Notropis calabazas*, *T. mandibularis*, *T. rasconis*, and *Xenoophorus captivus*, all of which are listed by the International Union for Conservation of Nature (IUCN) and NOM 059 SEMARNAT 2010 as critically endangered or threatened. Future expeditions to sites where they were previously collected should focus on searching for these species.

Two large groups formed in our analyses. One included non-native, invasive species whose ecological plasticity and life history traits (i.e., adaptability, high reproductive output, and dispersal capacity) allowed their expansion throughout the basin, potentially leading to negative interactions (i.e., interference, predation, and competition) with native species [88]. A second group was formed by native species, many of which are known from unique, often isolated, freshwater ecosystems (i.e., desert springs or small headwater streams) or have relatively small ranges (i.e., *Notropis calabazas*, *Xiphophorus pygmaeus*, *X. nigrensis*). Native faunas are generally vulnerable to the introduction of competitors and predators, in addition to being susceptible to disease [89]. That many native species were not registered in our latest time intervals suggests accelerated species loss. This needs to be addressed by the implementation of species-specific conservation measures. We acknowledge that our work is limited by a lack of concise information on species absence in databases, which prevents us from calculating the actual number of species disappearing in the Pánuco. However, 37% of species not found in the last time interval of our analysis are also listed by IUCN or NOM-059-SEMARNAT-2010, and another 35% are considered data-deficient by IUCN. Our study points to a potentially accelerated rate of species loss from a regional perspective. Further, we recognize that our datasets may be affected by differences in sampling efforts carried out over time. Our dataset results from the efforts from many scientists across decades whose sampling goals, strategies, and methods may have differed considerably. Large-scale studies that integrate information from a variety of sources are subject to such biases in sampling methodology [90–92]. Despite these limitations, we feel our use of rarefaction curves and relatively long time periods for analysis help address some of the issues derived from a lack of standardization [93].

### 4.3. Species Contribution

We identified 13 species generating significant changes across time intervals. Two were native and eleven (including five invasives) were non-native. Cyprinids *Cyprinus carpio*, *C. idella*, and *C. auratus* were introduced prior to the 1980s for use in aquaculture and aquariums and are today well established in the Pánuco and generally throughout the country [15,85]. Cichlids *A. nigrofasciata* and

*C. zilli*, also non-natives causing significant changes in across time intervals, are known for negatively affecting native cichlids and cyprinids [94–96]. Fourteen species were responsible for differences among sub-basins. Most of the species responsible for changes in among time-interval differences were also involved in among sub-basin differences. Only *Gambusia panuco* and *G. atrora* were responsible for among sub-basin differences, but not for among time interval differences. These results strongly suggest that non-native, invasive species of use in aquaculture could be largely responsible for changes in the fish composition of the Pánuco Basin. Local and federal programs for fomenting aquaculture will continue to be implemented in the country, leading to increased non-native species introductions [97] and affecting native fish communities [85]. Further threats include dewatering of many streams and rivers and pollution. For example, water irrigation districts such as Mante, Xocoténcatl, and Río Pánuco, Las Ánimas modify water courses and deviate steam water to sugar cane, citrus, and other crop plantations [98]. Especially in the upper (San Juan and Tula rivers) and lower (Tampico-Madero and Altamira rivers) Pánuco basin, pollution from agriculture and industry has seriously affected aquatic systems, rendering several river stretches uninhabitable [99]. These activities will continue to affect native faunas unless their operation is modified, taking into consideration needs for protecting natural habitats or via adoption of impact mitigation strategies, especially in areas of high endemism [100]. Our results can be used to identify species that are likely in the most danger of being affected by anthropogenic activities.

While much is known about how non-native species alter freshwater ecosystem structure and function, studies that incorporate a long-term and ample geographic scale perspective can help us understand the magnitude of the challenges imposed by biological homogenization and differentiation [101]. Being the first approximation to quantify fish fauna homogenization and differentiation rates in Mexico, we believe this study offers information and an analytical approach that could be implemented in other areas of the country. Further, it highlights aspects of faunal change that should be incorporated into national and regional strategies for biodiversity conservation.

## 5. Conclusions

The Pánuco River remains one with high ichthyological diversity despite being subject to numerous anthropogenic alterations over the last 40 years. Arrival of numerous non-native species and disappearance of endemic species has led to increasing ichthyological homogenization and then differentiation among the four sub-basins comprising the Pánuco Basin. Unfortunately, due to a lack of strict and effective conservation efforts, it is likely differentiation trends will continue as consequence of ongoing environmental degradation (primarily through damming and river desiccation). Homogenization might continue to occur as non-native species continue to expand throughout the basin, likely because of species dispersal via aquaculture programs and releases of aquarium species.

**Author Contributions:** N.M.-L. and N.M.-S. conceived the idea for this study with considerable assistance from J.J.S.-S.; N.M.-L and G.I.C.-R. collected data; N.M.-S., N.M.-L., and V.S.-D. developed various aspects of the methodology; N.M.-L and A.P.M.-F. carried out various analyses of the data; N.M.-S., N.M.-L., and H.M.-M. wrote, edited, and prepared the manuscript for submission.; All authors have read and agreed to the published version of the manuscript.

**Funding:** This research received no external funding.

**Acknowledgments:** We thank Ramírez-Herrejón J.P. for assistance in obtaining data from the "Edmundo Díaz Pardo" fish collection of the Universidad Autónoma de Querétaro. Octavio-Aguilar P. assisted with analyses. The Consejo Nacional de Ciencia y Tecnología awarded doctoral program support to N.M.-L. in calls-for-proposals no. 291197 and no. 291277. This paper is a product of N.M.-L.´s PhD degree at the Doctorado en Ciencias Naturales of the Universidad Autónoma del Estado de Morelos, Mexico.

**Conflicts of Interest:** The authors declare no conflict of interest.

# Appendix A

**Table A1.** Freshwater fish species from the Pánuco Basin registered (1 = presence, 0 = absence) in five time intervals and four sub-basins: (A) Río Pánuco, (B) Río Tamesí, (C) Río Tamuín, and (D) Río Moctezuma.

| No. | Scientific Name | Interval 1 (≤1980) | | | | Interval 2 (1981–1990) | | | | Interval 3 (1991–2000) | | | | Interval 4 (2001–2010) | | | | Interval 5 (2011–2018) | | | |
|---|---|---|---|---|---|---|---|---|---|---|---|---|---|---|---|---|---|---|---|---|---|
| | | A | B | C | D | A | B | C | D | A | B | C | D | A | B | C | D | A | B | C | D |
| 1 | *Algansea tincella* | 0 | 0 | 1 | 1 | 0 | 0 | 1 | 1 | 0 | 0 | 1 | 1 | 0 | 0 | 1 | 1 | 0 | 0 | 0 | 0 |
| 2 | *Amatitlania nigrofasciata* | 0 | 0 | 0 | 0 | 0 | 0 | 0 | 0 | 0 | 0 | 0 | 1 | 0 | 0 | 0 | 1 | 0 | 0 | 0 | 1 |
| 3 | *Astyanax rioverde* | 0 | 0 | 1 | 0 | 0 | 0 | 1 | 0 | 0 | 0 | 1 | 0 | 0 | 0 | 1 | 0 | 0 | 0 | 1 | 0 |
| 4 | *Astyanax mexicanus* | 1 | 1 | 1 | 1 | 1 | 1 | 1 | 1 | 1 | 1 | 1 | 1 | 1 | 1 | 1 | 1 | 1 | 1 | 1 | 1 |
| 5 | *Ataeniobius toweri* | 0 | 0 | 1 | 0 | 0 | 0 | 1 | 0 | 0 | 0 | 1 | 0 | 0 | 0 | 1 | 0 | 0 | 0 | 0 | 0 |
| 6 | *Atractosteus spatula* | 1 | 1 | 1 | 1 | 0 | 0 | 0 | 0 | 0 | 0 | 0 | 0 | 0 | 0 | 0 | 0 | 0 | 0 | 0 | 0 |
| 7 | *Carassius auratus* | 0 | 1 | 0 | 1 | 0 | 0 | 0 | 1 | 0 | 0 | 0 | 1 | 0 | 0 | 1 | 1 | 0 | 0 | 1 | 1 |
| 8 | *Chirostoma arge* | 0 | 0 | 0 | 0 | 0 | 0 | 0 | 0 | 0 | 0 | 0 | 0 | 0 | 0 | 0 | 0 | 0 | 0 | 1 | 0 |
| 9 | *Chirostoma grandocule* | 1 | 0 | 0 | 1 | 0 | 0 | 0 | 0 | 0 | 0 | 0 | 0 | 0 | 0 | 0 | 0 | 0 | 0 | 0 | 0 |
| 10 | *Chirostoma humboldtianum* | 0 | 0 | 0 | 1 | 0 | 0 | 0 | 1 | 0 | 0 | 0 | 1 | 0 | 0 | 0 | 1 | 0 | 0 | 0 | 0 |
| 11 | *Chirostoma jordani* | 0 | 0 | 0 | 1 | 0 | 0 | 0 | 1 | 0 | 0 | 0 | 1 | 0 | 0 | 0 | 1 | 0 | 0 | 0 | 0 |
| 12 | *Coptodon rendalli* | 0 | 0 | 0 | 1 | 0 | 0 | 0 | 0 | 0 | 0 | 0 | 0 | 0 | 0 | 0 | 0 | 0 | 0 | 0 | 0 |
| 13 | *Coptodon zillii* | 0 | 1 | 0 | 1 | 0 | 0 | 0 | 0 | 0 | 0 | 0 | 0 | 0 | 0 | 0 | 0 | 0 | 0 | 0 | 0 |
| 14 | *Ctenopharyngodon idella* | 0 | 0 | 0 | 0 | 0 | 0 | 0 | 0 | 0 | 0 | 0 | 0 | 0 | 1 | 1 | 0 | 0 | 0 | 0 | 0 |
| 15 | *Cualac tessellatus* | 0 | 0 | 1 | 0 | 0 | 0 | 1 | 0 | 0 | 0 | 1 | 0 | 0 | 0 | 1 | 0 | 0 | 0 | 0 | 0 |
| 16 | *Cyprinella lutrensis* | 0 | 1 | 1 | 1 | 0 | 0 | 0 | 1 | 0 | 0 | 0 | 1 | 0 | 0 | 0 | 1 | 0 | 0 | 0 | 0 |
| 17 | *Cyprinus carpio* | 0 | 0 | 0 | 1 | 0 | 0 | 0 | 1 | 0 | 0 | 0 | 1 | 0 | 1 | 0 | 1 | 0 | 1 | 1 | 1 |
| 18 | *Gambusia affinis* | 1 | 1 | 1 | 1 | 0 | 1 | 1 | 1 | 0 | 1 | 1 | 1 | 0 | 1 | 1 | 1 | 0 | 1 | 1 | 0 |
| 19 | *Gambusia atrora* | 0 | 0 | 0 | 1 | 0 | 0 | 0 | 1 | 0 | 0 | 0 | 1 | 0 | 0 | 0 | 1 | 1 | 0 | 0 | 0 |
| 20 | *Gambusia aurata* | 0 | 1 | 0 | 0 | 0 | 1 | 0 | 0 | 0 | 1 | 0 | 0 | 0 | 1 | 0 | 0 | 0 | 0 | 0 | 0 |
| 21 | *Gambusia marshi* | 0 | 1 | 0 | 0 | 0 | 1 | 0 | 0 | 0 | 1 | 0 | 0 | 1 | 0 | 0 | 0 | 0 | 0 | 0 | 0 |
| 22 | *Gambusia panuco* | 1 | 1 | 1 | 1 | 0 | 1 | 1 | 1 | 0 | 1 | 1 | 1 | 0 | 1 | 1 | 0 | 0 | 0 | 1 | 0 |
| 23 | *Gambusia quadruncus* | 1 | 0 | 1 | 1 | 0 | 0 | 0 | 0 | 0 | 0 | 0 | 0 | 0 | 0 | 0 | 0 | 0 | 0 | 0 | 0 |
| 24 | *Gambusia regani* | 1 | 1 | 1 | 1 | 1 | 1 | 1 | 1 | 1 | 1 | 1 | 1 | 1 | 1 | 1 | 1 | 0 | 0 | 1 | 1 |
| 25 | *Gambusia vittata* | 1 | 1 | 1 | 1 | 1 | 1 | 1 | 1 | 1 | 1 | 1 | 1 | 1 | 1 | 1 | 1 | 0 | 1 | 0 | 1 |
| 26 | *Girardinichthys multiradiatus* | 0 | 0 | 0 | 1 | 0 | 0 | 0 | 0 | 0 | 0 | 0 | 0 | 0 | 0 | 0 | 0 | 0 | 0 | 0 | 0 |
| 27 | *Girardinichthys viviparus* | 0 | 0 | 0 | 1 | 0 | 0 | 0 | 1 | 0 | 0 | 0 | 1 | 0 | 0 | 0 | 1 | 0 | 0 | 0 | 1 |
| 28 | *Goodea atripinnis* | 0 | 0 | 1 | 1 | 0 | 0 | 1 | 1 | 0 | 0 | 1 | 1 | 0 | 0 | 1 | 1 | 0 | 0 | 1 | 1 |
| 29 | *Goodea gracilis* | 0 | 0 | 1 | 1 | 0 | 0 | 1 | 1 | 0 | 0 | 1 | 1 | 0 | 0 | 1 | 1 | 0 | 0 | 0 | 1 |
| 30 | *Herichthys bartoni* | 0 | 0 | 1 | 0 | 0 | 0 | 1 | 0 | 0 | 0 | 1 | 0 | 0 | 0 | 1 | 0 | 0 | 0 | 0 | 0 |
| 31 | *Herichthys carpintis* | 1 | 1 | 1 | 1 | 0 | 0 | 1 | 1 | 0 | 0 | 1 | 1 | 0 | 0 | 1 | 0 | 0 | 0 | 1 | 0 |
| 32 | *Herichthys cyanoguttatus* | 1 | 1 | 1 | 1 | 1 | 1 | 1 | 1 | 1 | 1 | 1 | 1 | 1 | 1 | 1 | 1 | 0 | 0 | 1 | 1 |
| 33 | *Herichthys labridens* | 0 | 1 | 1 | 0 | 0 | 1 | 1 | 0 | 0 | 1 | 1 | 0 | 0 | 1 | 1 | 0 | 0 | 0 | 1 | 0 |
| 34 | *Herichthys molango* | 0 | 0 | 0 | 1 | 0 | 0 | 0 | 1 | 0 | 0 | 0 | 1 | 0 | 0 | 0 | 1 | 0 | 0 | 0 | 0 |
| 35 | *Herichthys pame* | 0 | 0 | 1 | 0 | 0 | 0 | 1 | 0 | 0 | 0 | 1 | 0 | 0 | 0 | 0 | 0 | 0 | 0 | 0 | 0 |
| 36 | *Herichthys pantostictus* | 1 | 1 | 1 | 1 | 1 | 1 | 1 | 1 | 1 | 1 | 1 | 1 | 1 | 1 | 1 | 1 | 0 | 0 | 1 | 0 |
| 37 | *Herichthys steindachneri* | 0 | 1 | 1 | 0 | 0 | 1 | 1 | 0 | 0 | 1 | 1 | 0 | 0 | 1 | 1 | 1 | 0 | 0 | 1 | 0 |
| 38 | *Herichthys tamasopoensis* | 0 | 0 | 1 | 0 | 0 | 0 | 1 | 0 | 0 | 0 | 1 | 0 | 0 | 0 | 1 | 0 | 0 | 0 | 1 | 0 |
| 39 | *Hypophthalmichthys molitrix* | 0 | 0 | 0 | 0 | 0 | 0 | 0 | 0 | 0 | 0 | 0 | 0 | 0 | 1 | 0 | 0 | 0 | 1 | 0 | 1 |
| 40 | *Ictalurus australis* | 0 | 1 | 1 | 1 | 0 | 1 | 1 | 1 | 0 | 1 | 1 | 1 | 0 | 1 | 1 | 1 | 0 | 0 | 0 | 0 |
| 41 | *Ictalurus furcatus* | 1 | 1 | 1 | 1 | 0 | 1 | 0 | 0 | 0 | 1 | 0 | 0 | 0 | 0 | 0 | 0 | 0 | 1 | 0 | 0 |
| 42 | *Ictalurus lupus* | 0 | 1 | 0 | 0 | 0 | 0 | 0 | 0 | 0 | 0 | 0 | 0 | 0 | 0 | 0 | 0 | 0 | 0 | 0 | 0 |
| 43 | *Ictalurus mexicanus* | 0 | 1 | 1 | 1 | 0 | 1 | 1 | 1 | 0 | 1 | 1 | 1 | 0 | 1 | 1 | 1 | 0 | 0 | 1 | 0 |
| 44 | *Ictalurus punctatus* | 0 | 1 | 1 | 1 | 0 | 1 | 0 | 1 | 0 | 1 | 0 | 1 | 0 | 1 | 0 | 1 | 0 | 1 | 0 | 1 |
| 45 | *Ictiobus bubalus* | 0 | 1 | 0 | 0 | 0 | 1 | 0 | 0 | 0 | 0 | 1 | 0 | 0 | 0 | 0 | 0 | 0 | 0 | 0 | 0 |
| 46 | *Ictiobus labiosus* | 1 | 1 | 1 | 1 | 0 | 0 | 1 | 1 | 0 | 0 | 1 | 1 | 0 | 0 | 1 | 1 | 0 | 0 | 1 | 0 |
| 47 | *Lepisosteus osseus* | 1 | 1 | 1 | 1 | 0 | 0 | 0 | 1 | 0 | 0 | 0 | 1 | 0 | 0 | 0 | 1 | 0 | 0 | 0 | 0 |
| 48 | *Lepomis macrochirus* | 0 | 0 | 0 | 1 | 0 | 0 | 0 | 1 | 0 | 0 | 0 | 1 | 0 | 0 | 0 | 1 | 0 | 0 | 1 | 1 |
| 49 | *Micropterus salmoides* | 1 | 0 | 1 | 1 | 1 | 0 | 1 | 1 | 1 | 0 | 1 | 1 | 1 | 1 | 1 | 1 | 1 | 1 | 1 | 1 |
| 50 | *Notropis aguirrepequenoi* | 0 | 1 | 0 | 0 | 0 | 0 | 0 | 0 | 0 | 0 | 0 | 0 | 0 | 0 | 0 | 0 | 0 | 0 | 0 | 0 |
| 51 | *Notropis boucardi* | 0 | 0 | 0 | 0 | 0 | 0 | 0 | 0 | 0 | 0 | 1 | 0 | 0 | 0 | 0 | 0 | 0 | 0 | 0 | 0 |
| 52 | *Notropis calabazas* | 0 | 0 | 1 | 0 | 0 | 0 | 1 | 0 | 0 | 0 | 1 | 0 | 0 | 0 | 1 | 0 | 0 | 0 | 0 | 0 |
| 53 | *Notropis calientis* | 0 | 1 | 1 | 0 | 0 | 0 | 0 | 0 | 0 | 0 | 0 | 0 | 0 | 0 | 0 | 0 | 0 | 0 | 0 | 0 |
| 54 | *Notropis sallaei* | 0 | 0 | 1 | 1 | 0 | 0 | 0 | 1 | 0 | 0 | 0 | 1 | 0 | 0 | 1 | 0 | 0 | 0 | 0 | 0 |
| 55 | *Notropis tropicus* | 0 | 1 | 1 | 1 | 0 | 1 | 0 | 1 | 0 | 1 | 0 | 1 | 0 | 1 | 0 | 1 | 0 | 0 | 0 | 0 |
| 56 | *Oncorhynchus mykiss* | 0 | 0 | 0 | 0 | 0 | 0 | 0 | 0 | 0 | 0 | 1 | 0 | 0 | 0 | 0 | 1 | 0 | 0 | 0 | 1 |
| 57 | *Oreochromis aureus* | 0 | 0 | 0 | 1 | 0 | 0 | 0 | 1 | 0 | 0 | 1 | 1 | 1 | 1 | 1 | 1 | 1 | 1 | 1 | 1 |
| 58 | *Oreochromis mossambicus* | 0 | 0 | 0 | 1 | 0 | 0 | 0 | 1 | 0 | 0 | 1 | 1 | 0 | 1 | 1 | 1 | 0 | 1 | 1 | 1 |
| 59 | *Oreochromis niloticus* | 0 | 0 | 0 | 0 | 0 | 0 | 0 | 0 | 0 | 0 | 1 | 1 | 0 | 1 | 1 | 1 | 0 | 1 | 1 | 1 |
| 60 | *Poecilia formosa* | 1 | 1 | 1 | 1 | 1 | 1 | 0 | 0 | 1 | 1 | 0 | 0 | 1 | 1 | 0 | 0 | 0 | 0 | 0 | 0 |
| 61 | *Poecilia latipinna* | 1 | 1 | 0 | 0 | 0 | 0 | 0 | 0 | 0 | 0 | 0 | 0 | 0 | 0 | 0 | 0 | 0 | 0 | 0 | 0 |
| 62 | *Poecilia latipunctata* | 0 | 1 | 1 | 1 | 0 | 1 | 1 | 1 | 0 | 1 | 1 | 1 | 0 | 1 | 1 | 0 | 0 | 0 | 0 | 0 |
| 63 | *Poecilia limantouri* | 0 | 1 | 1 | 1 | 0 | 0 | 0 | 0 | 0 | 0 | 0 | 0 | 0 | 0 | 0 | 0 | 0 | 0 | 0 | 0 |

**Table A1.** *Cont.*

| No. | Scientific Name | Interval 1 (≤1980) | | | | Interval 2 (1981–1990) | | | | Interval 3 (1991–2000) | | | | Interval 4 (2001–2010) | | | | Interval 5 (2011–2018) | | | |
|---|---|---|---|---|---|---|---|---|---|---|---|---|---|---|---|---|---|---|---|---|---|
| | | A | B | C | D | A | B | C | D | A | B | C | D | A | B | C | D | A | B | C | D |
| 64 | *Poecilia mexicana* | 1 | 1 | 1 | 1 | 1 | 1 | 1 | 1 | 1 | 1 | 1 | 1 | 1 | 1 | 1 | 1 | 0 | 0 | 1 | 1 |
| 65 | *Poecilia reticulata* | 0 | 0 | 0 | 1 | 0 | 0 | 0 | 1 | 0 | 0 | 0 | 1 | 0 | 0 | 0 | 1 | 0 | 0 | 0 | 1 |
| 66 | *Poecilia sphenops* | 0 | 1 | 1 | 0 | 0 | 0 | 1 | 0 | 0 | 0 | 1 | 0 | 0 | 0 | 0 | 0 | 0 | 0 | 0 | 0 |
| 67 | *Poeciliopsis gracilis* | 0 | 1 | 1 | 1 | 0 | 0 | 1 | 1 | 0 | 0 | 1 | 1 | 1 | 0 | 1 | 1 | 0 | 0 | 1 | 0 |
| 68 | *Poeciliopsis infans* | 0 | 0 | 0 | 1 | 0 | 0 | 0 | 0 | 0 | 0 | 1 | 0 | 0 | 0 | 1 | 0 | 0 | 0 | 1 | 0 |
| 69 | *Poeciliopsis lucida* | 0 | 0 | 0 | 0 | 0 | 0 | 0 | 0 | 0 | 0 | 1 | 0 | 0 | 0 | 0 | 0 | 0 | 0 | 0 | 0 |
| 70 | *Pseudoxiphophorus bimaculatus* | 0 | 0 | 0 | 1 | 0 | 0 | 0 | 1 | 0 | 0 | 1 | 1 | 0 | 0 | 1 | 1 | 0 | 0 | 1 | 1 |
| 71 | *Pseudoxiphophorus jonesii* | 0 | 0 | 1 | 1 | 0 | 0 | 1 | 1 | 0 | 0 | 1 | 1 | 0 | 0 | 1 | 1 | 0 | 0 | 0 | 1 |
| 72 | *Pylodictis olivaris* | 0 | 0 | 1 | 1 | 0 | 0 | 0 | 0 | 0 | 0 | 0 | 0 | 0 | 1 | 0 | 0 | 0 | 0 | 0 | 0 |
| 73 | *Tampichthys catostomops* | 0 | 0 | 1 | 0 | 0 | 0 | 1 | 0 | 0 | 0 | 1 | 0 | 0 | 0 | 1 | 0 | 0 | 0 | 1 | 0 |
| 74 | *Tampichthys dichroma* | 0 | 0 | 1 | 0 | 0 | 0 | 1 | 0 | 0 | 0 | 1 | 0 | 0 | 0 | 1 | 0 | 0 | 0 | 1 | 0 |
| 75 | *Tampichthys erimyzonops* | 0 | 1 | 1 | 1 | 0 | 1 | 0 | 1 | 0 | 1 | 0 | 1 | 0 | 1 | 0 | 1 | 0 | 0 | 0 | 0 |
| 76 | *Tampichthys ipni* | 0 | 1 | 1 | 1 | 0 | 1 | 1 | 1 | 0 | 1 | 1 | 1 | 0 | 1 | 0 | 1 | 0 | 0 | 0 | 0 |
| 77 | *Tampichthys mandibularis* | 0 | 0 | 1 | 0 | 0 | 0 | 1 | 0 | 0 | 0 | 1 | 0 | 0 | 0 | 1 | 0 | 0 | 0 | 0 | 0 |
| 78 | *Tampichthys rasconis* | 0 | 0 | 1 | 0 | 0 | 0 | 1 | 0 | 0 | 0 | 1 | 0 | 0 | 0 | 0 | 0 | 0 | 0 | 0 | 0 |
| 79 | *Thorichthys maculipinnis* | 0 | 0 | 0 | 0 | 0 | 0 | 0 | 0 | 0 | 0 | 0 | 0 | 0 | 0 | 1 | 0 | 0 | 0 | 0 | 0 |
| 80 | *Xenoophorus captivus* | 0 | 0 | 1 | 1 | 0 | 0 | 1 | 0 | 0 | 0 | 1 | 0 | 0 | 0 | 1 | 0 | 0 | 0 | 0 | 0 |
| 81 | *Xenotoca variata* | 0 | 0 | 1 | 1 | 0 | 0 | 1 | 1 | 0 | 0 | 1 | 1 | 0 | 0 | 1 | 0 | 0 | 0 | 1 | 0 |
| 82 | *Xiphophorus birchmanni* | 0 | 0 | 0 | 1 | 0 | 0 | 0 | 1 | 0 | 0 | 0 | 1 | 1 | 0 | 0 | 1 | 0 | 0 | 0 | 1 |
| 83 | *Xiphophorus continens* | 0 | 0 | 1 | 0 | 0 | 0 | 1 | 0 | 0 | 0 | 0 | 0 | 0 | 0 | 0 | 0 | 0 | 0 | 0 | 0 |
| 84 | *Xiphophorus cortezi* | 0 | 0 | 1 | 1 | 0 | 0 | 1 | 1 | 0 | 0 | 1 | 1 | 0 | 0 | 1 | 1 | 0 | 0 | 0 | 0 |
| 85 | *Xiphophorus hellerii* | 0 | 0 | 0 | 1 | 0 | 0 | 0 | 1 | 0 | 0 | 0 | 1 | 0 | 0 | 0 | 1 | 0 | 0 | 0 | 0 |
| 86 | *Xiphophorus maculatus* | 0 | 1 | 0 | 0 | 0 | 0 | 0 | 0 | 0 | 0 | 0 | 0 | 0 | 0 | 0 | 0 | 0 | 0 | 0 | 0 |
| 87 | *Xiphophorus malinche* | 0 | 0 | 0 | 1 | 0 | 0 | 0 | 1 | 0 | 0 | 0 | 1 | 0 | 0 | 0 | 1 | 0 | 0 | 0 | 1 |
| 88 | *Xiphophorus montezumae* | 0 | 1 | 1 | 0 | 0 | 0 | 1 | 1 | 0 | 0 | 1 | 0 | 0 | 0 | 1 | 0 | 0 | 0 | 1 | 0 |
| 89 | *Xiphophorus multilineatus* | 0 | 0 | 1 | 0 | 0 | 0 | 1 | 0 | 0 | 0 | 1 | 0 | 0 | 0 | 1 | 0 | 0 | 0 | 1 | 0 |
| 90 | *Xiphophorus nezahualcoyotl* | 0 | 1 | 1 | 0 | 0 | 1 | 1 | 0 | 0 | 1 | 1 | 0 | 0 | 1 | 1 | 0 | 0 | 0 | 0 | 0 |
| 91 | *Xiphophorus nigrensis* | 0 | 0 | 1 | 0 | 0 | 0 | 1 | 0 | 0 | 0 | 1 | 0 | 0 | 0 | 0 | 0 | 0 | 0 | 0 | 0 |
| 92 | *Xiphophorus pygmaeus* | 0 | 0 | 1 | 1 | 0 | 0 | 0 | 1 | 0 | 0 | 0 | 1 | 0 | 1 | 0 | 1 | 0 | 0 | 0 | 0 |
| 93 | *Xiphophorus variatus* | 1 | 1 | 1 | 1 | 1 | 1 | 1 | 1 | 1 | 1 | 1 | 1 | 1 | 1 | 1 | 1 | 0 | 1 | 1 | 1 |
| 94 | *Xiphophorus xiphidium* | 0 | 1 | 0 | 0 | 0 | 0 | 0 | 0 | 0 | 0 | 0 | 0 | 0 | 0 | 0 | 0 | 0 | 0 | 0 | 0 |
| 95 | *Yuriria alta* | 0 | 0 | 0 | 0 | 0 | 0 | 0 | 0 | 0 | 0 | 0 | 0 | 0 | 0 | 0 | 0 | 0 | 0 | 1 | 0 |

**Table A2.** Categories of freshwater fish species from the Pánuco Basin. We include origin for species, N = native to the basin, NN = non native to the basin; if NN, we specify which sub-basin they are NN in (A = Pánuco, B = Tamesí, C = Tamuín, and D = Moctezuma). We also include species categorized as invasive by Comisión Nacional para el Conocimiento y Uso de la Biodiversidad (CONABIO) and the category for species under the International Union for Conservation of Nature's Red List of Threatened Species and the Norma Oficial Mexicana NOM 059 SEMARNAT 2010 (see below).

| No. | Family | Scientific Name | ORIGIN | | Invasive (Yes /No) | Red List Iucn | Nom 059 Semarnat 2010 |
|---|---|---|---|---|---|---|---|
| | | | (Native = N, Non Native = NN) | Sub-Basin (A,B,C,D) | | | |
| 1 | Cyprinidae | *Algansea tincella* | N | C | No | LC | No |
| 2 | Cichlidae | *Amatitlania nigrofasciata* | NN | | Yes | NE | No |
| 3 | Characidae | *Astyanax rioverde* | N | C | No | LC | No |
| 4 | Characidae | *Astyanax mexicanus* * | N | ABCD | No | LC/VU | No/A |
| 5 | Goodeidae | *Ataeniobius toweri* | N | C | No | EN | P |
| 6 | Lepisosteidae | *Atractosteus spatula* | N | B | No | LC | No |
| 7 | Cyprinidae | *Carassius auratus* | NN | | Yes | LC | No |
| 8 | Atherinopsidae | *Chirostoma arge* | NN | | No | DD | No |
| 9 | Atherinopsidae | *Chirostoma grandocule* | NN | | No | DD | No |
| 10 | Atherinopsidae | *Chirostoma humboldtianum* | NN | | No | NE | No |
| 11 | Atherinopsidae | *Chirostoma jordani* | NN | | No | LC | No |
| 12 | Cichlidae | *Coptodon rendalli* | NN | | No | LC | No |
| 13 | Cichlidae | *Coptodon zillii* | NN | | Yes | LC | No |
| 14 | Cyprinidae | *Ctenopharyngodon idella* | NN | | Yes | NE | No |
| 15 | Cyprinodontidae | *Cualac tessellatus* | N | C | No | VU | P |
| 16 | Cyprinidae | *Cyprinella lutrensis* | N | C | No | LC | A |
| 17 | Cyprinidae | *Cyprinus carpio* | NN | | Yes | VU | No |
| 18 | Poeciliidae | *Gambusia affinis* | NN | | No | LC | No |
| 19 | Poeciliidae | *Gambusia atrora* | N | D | No | DD | No |

**Table A2.** *Cont.*

| No. | Family | Scientific Name | ORIGIN | | Invasive (Yes /No) | Red List Iucn | Nom 059 Semarnat 2010 |
|---|---|---|---|---|---|---|---|
| | | | (Native = N, Non Native = NN) | Sub-Basin (A,B,C,D) | | | |
| 20 | Poeciliidae | *Gambusia aurata* | N | B | No | DD | No |
| 21 | Poeciliidae | *Gambusia marshi* | NN | | No | LC | A |
| 22 | Poeciliidae | *Gambusia panuco* | N | AB | No | DD | No |
| 23 | Poeciliidae | *Gambusia quadruncus* | N | AC | No | NE | No |
| 24 | Poeciliidae | *Gambusia regani* | N | ABCD | No | DD | No |
| 25 | Poeciliidae | *Gambusia vittata* | N | B | No | LC | No |
| 26 | Goodeidae | *Girardinichthys multiradiatus* | NN | | No | EN | No |
| 27 | Goodeidae | *Girardinichthys viviparus* | NN | | No | EN | P |
| 28 | Goodeidae | *Goodea atripinnis* | N | C | No | LC | No |
| 29 | Goodeidae | *Goodea gracilis* | N | C | No | LC | No |
| 30 | Cichlidae | *Herichthys bartoni* | N | C | No | EN | P |
| 31 | Cichlidae | *Herichthys carpintis* | N | AB | No | LC | No |
| 32 | Cichlidae | *Herichthys cyanoguttatus* | NN | | No | LC | No |
| 33 | Cichlidae | *Herichthys labridens* | N | B | No | EN | A |
| 34 | Cichlidae | *Herichthys molango* | N | D | No | LC | No |
| 35 | Cichlidae | *Herichthys pame* | N | C | No | NE | No |
| 36 | Cichlidae | *Herichthys pantostictus* | N | BA | No | LC | No |
| 37 | Cichlidae | *Herichthys steindachneri* | N | C | No | EN | P |
| 38 | Cichlidae | *Herichthys tamasopoensis* | N | C | No | VU | No |
| 39 | Cyprinidae | *Hypophthalmichthys molitrix* | NN | | Yes | NT | No |
| 40 | Ictaluridae | *Ictalurus australis* | N | B | No | DD | A |
| 41 | Ictaluridae | *Ictalurus furcatus* | N | CD | No | LC | No |
| 42 | Ictaluridae | *Ictalurus lupus* | NN | | No | DD | No |
| 43 | Ictaluridae | *Ictalurus mexicanus* | N | C | No | VU | A |
| 44 | Ictaluridae | *Ictalurus punctatus* | N | BCD | No | LC | No |
| 45 | Catostomidae | *Ictiobus bubalus* | NN | | No | LC | A |
| 46 | Catostomidae | *Ictiobus labiosus* | N | CD | No | DD | No |
| 47 | Lepisosteidae | *Lepisosteus osseus* | N | ABC | No | LC | No |
| 48 | Centrarchidae | *Lepomis macrochirus* | NN | | No | LC | No |
| 49 | Centrarchidae | *Micropterus salmoides* | NN | | Yes | LC | No |
| 50 | Cyprinidae | *Notropis aguirrepequenoi* | NN | | No | VU | Pr |
| 51 | Cyprinidae | *Notropis boucardi* | NN | | No | EN | A |
| 52 | Cyprinidae | *Notropis calabazas* | N | C | No | CR | No |
| 53 | Cyprinidae | *Notropis calientis* | N | C | No | CR | No |
| 54 | Cyprinidae | *Notropis sallaei* | N | D | No | LC | No |
| 55 | Cyprinidae | *Notropis tropicus* | N | BC | No | NT | No |
| 56 | Salmonidae | *Oncorhynchus mykiss* | NN | | Yes | NE | Pr |
| 57 | Cichlidae | *Oreochromis aureus* | NN | | Yes | LC | No |
| 58 | Cichlidae | *Oreochromis mossambicus* | NN | | Yes | NT | No |
| 59 | Cichlidae | *Oreochromis niloticus* | NN | | Yes | LC | No |
| 60 | Poeciliidae | *Poecilia formosa* | N | ABD | No | LC | No |
| 61 | Poeciliidae | *Poecilia latipinna* | NN | | No | LC | No |
| 62 | Poeciliidae | *Poecilia latipunctata* | N | B | No | DD | P |
| 63 | Poeciliidae | *Poecilia limantouri* | NN | | No | NE | No |
| 64 | Poeciliidae | *Poecilia mexicana* | N | ABCD | No | LC | No |
| 65 | Poeciliidae | *Poecilia reticulata* | NN | | Yes | NE | No |
| 66 | Poeciliidae | *Poecilia sphenops* | NN | | No | LC | Pr |
| 67 | Poeciliidae | *Poeciliopsis gracilis* | NN | | No | LC | No |
| 68 | Poeciliidae | *Poeciliopsis infans* | NN | | No | LC | No |
| 69 | Poeciliidae | *Poeciliopsis lucida* | NN | | No | DD | No |
| 70 | Poeciliidae | *Pseudoxiphophorus bimaculatus* | NN | | No | LC | No |
| 71 | Poeciliidae | *Pseudoxiphophorus jonesii* | NN | | No | LC | No |
| 72 | Ictaluridae | *Pylodictis olivaris* | NN | | No | LC | No |
| 73 | Cyprinidae | *Tampichthys catostomops* | N | C | No | NT | No |
| 74 | Cyprinidae | *Tampichthys dichroma* | N | C | No | NE | A |
| 75 | Cyprinidae | *Tampichthys erimyzonops* | N | BC | No | DD | No |
| 76 | Cyprinidae | *Tampichthys ipni* | N | BCD | No | LC | No |
| 77 | Cyprinidae | *Tampichthys mandibularis* | N | C | No | EN | P |
| 78 | Cyprinidae | *Tampichthys rasconis* | N | C | No | EN | No |
| 79 | Cichlidae | *Thorichthys maculipinnis* | NN | | No | NE | No |
| 80 | Goodeidae | *Xenoophorus captivus* | N | C | No | EN | P |
| 81 | Goodeidae | *Xenotoca variata* | N | C | No | LC | No |
| 82 | Poeciliidae | *Xiphophorus birchmanni* | N | D | No | LC | No |
| 83 | Poeciliidae | *Xiphophorus continens* | N | C | No | DD | No |
| 84 | Poeciliidae | *Xiphophorus cortezi* | N | CD | No | DD | No |
| 85 | Poeciliidae | *Xiphophorus hellerii* | NN | | No | LC | No |

**Table A2.** *Cont.*

| No. | Family | Scientific Name | ORIGIN | | Invasive (Yes /No) | Red List Iucn | Nom 059 Semarnat 2010 |
|---|---|---|---|---|---|---|---|
| | | | (Native = N, Non Native = NN) | Sub-Basin (A,B,C,D) | | | |
| 86 | Poeciliidae | *Xiphophorus maculatus* | NN | | No | DD | No |
| 87 | Poeciliidae | *Xiphophorus malinche* | N | D | No | DD | No |
| 88 | Poeciliidae | *Xiphophorus montezumae* | N | C | No | DD | No |
| 89 | Poeciliidae | *Xiphophorus multilineatus* | N | C | No | DD | No |
| 90 | Poeciliidae | *Xiphophorus nezahualcoyotl* | N | BC | No | DD | No |
| 91 | Poeciliidae | *Xiphophorus nigrensis* | N | C | No | DD | No |
| 92 | Poeciliidae | *Xiphophorus pygmaeus* | N | C | No | DD | No |
| 93 | Poeciliidae | *Xiphophorus variatus* | N | ABCD | No | LC | No |
| 94 | Poeciliidae | *Xiphophorus xiphidium* | NN | | No | LC | No |
| 95 | Cyprinidae | *Yuriria alta* | NN | | No | EN | No |

RED LIST IUCN—the International Union for Conservation of Nature's Red List of Threatened Species: NE = not evaluated; DD = data deficient; LC = least concern; NT = near threatened; VU = vulnerable; EN = endangered; CR = critically endangered. NOM 059 SEMARNAT 2010—NORMA Oficial Mexicana NOM-059-SEMARNAT-2010, Protección ambiental-Especies nativas de México de flora y fauna silvestres-Categorías de riesgo y especificaciones para su inclusión, exclusión o cambio-Lista de especies en riesgo: No = not in the NOM; Pr = subject to special protection; A = amenazadas (threatened); P = en peligro de extinción (in danger of extinction).* For *A. mexicanus*, the blind form was clasified as vulnerable (VU) by IUCN and (A) in NOM-059 SEMARNAT 2010 [102,103].

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
