# Peer review of "Ichthyological Differentiation and Homogenization in the Pánuco Basin, Mexico"

_diversity, doi:10.3390/d12050187_

Round 1

Reviewer 1 Report

Overall this is an interesting paper on a diverse freshwater system in which the authors utilise existing data on freshwater fish occurrences from the Pánuco Basin in Mexico to understand if there has been changes to species composition. The authors point to some interesting findings regarding non-native fishes and also those that are of conservation concern. 

Although the authors do state the limitations of the dataset at the end of their manuscript, I feel that they could be more explicit about the limitations. For example, are collecting efforts greater now than earlier on? or is the reverse true. Have collecting efforts been uniform across taxonomic groups? across basins etc? 

Ln 99 – what are primary and secondary species – define…

Reviewer 2 Report

The manuscript entitled “Ichthyological differentiation and homogenization in 2 the Pánuco Basin, Mexico” is very interesting and correctly written. The problem raised by the authors is very important and current all over the world.

TABLES

Please improve the Table 1, 2 and 3  according to the requirements of the Water magazine

Table 1 - Please explain the codes used e.g. Sps –Species?

FIGURES

Figure 2. The font is too big, for example: “Time intervals” or “Sub-basins”.

REFERENCES

In the text, reference numbers should be placed before the punctuation; for example [1], [1–3] or [1,3]. LINE 39 [2-5] ; LINE 45 [8,9,10]  should be [8-10].

Please check the correctness of writing references.

Detailed Comments

Line 160 – Please explain the abbreviation- UTM, WGS_1984

Line 39, 131, 136, 272……..  – Please put a space between words

Line 45- There is no dot after sentence.

Line 79-  What does “y” mean?
